# sNet: Not All Edges Matter Equally in Temporal Link Prediction

## Abstract

Temporal link prediction, aimed at forecasting future interactions from historical ones, is fundamental to understanding graph evolution and supports a wide range of practical applications. Despite recent progress, scalability remains a major concern: the typically adopted per-query likelihood estimation requires a series of costly operations (e.g., relative encodings and historical neighbor sampling) for each query link, resulting in prohibitive time costs when the number of query links is large. By analyzing the state-of-the-art temporal link prediction method on the TGB leaderboard, we identify that converting maintained node memories into edge embeddings dominates the computational cost, accounting for over $90\%$ of the runtime. Surprisingly, this operation is unnecessary for most queries, where applying a simple threshold on memory-based predictions can filter out about $80\%$ of edges with negligible loss in accuracy. Motivated by this, we propose a neural selector plug-in called sNet, which enables a memory-based method to adaptively choose between memory and embedding predictions. Specifically, sNet first outputs computationally cheap memory predictions, and then refines the unreliable predictions into embedding predictions based on a neural selector. A tailored surrogate loss is introduced to train the non-differentiable selection process, together with a dynamic weight adjustment strategy that automatically tunes the balance between memory and embedding predictions towards the preferred performance threshold, thereby reducing reliance on trial-and-error tuning. Experimental results on the TGB benchmark demonstrate the effectiveness of the proposed method, with sNet enabling the SOTA method to achieve an average $5.56\times$ speedup while incurring only a $0.69\%$ performance drop.

## 1 Introduction

From online social networks to financial trading systems, temporal graphs provide a powerful abstraction for ubiquitous interconnected and dynamically evolving data (Holme & Saramäki, 2012), where entities are represented as nodes and their interactions as timestamped edges. Temporal link prediction, which aims to forecast future interactions from historical ones, is not only essential for understanding the evolution of real-world dynamic systems but also underpins a wide range of applications such as recommendation (Fan et al., 2021; Kumar et al., 2019), fraud detection (Cao et al., 2025; Huang et al., 2022), and information diffusion prediction (Lu et al., 2023).

Despite recent progress, scalability remains a major concern for temporal link prediction methods. Specifically, for each query link, these methods typically rely on a computationally intensive likelihood estimation mechanism, often involving operations such as historical neighbor sampling (Rossi et al., 2020; Lu et al., 2025) and relative encoding (Wang et al., 2021; Lu et al., 2024). While effective, this design incurs prohibitive overhead as the number of query links grows. As shown in Figure 1, applying the same method on the same dataset under different benchmark settings (DyGLib vs. TGB) can result in runtime differences of up to **60×**, driven solely by the number of negative samples. Such poor scalabil-

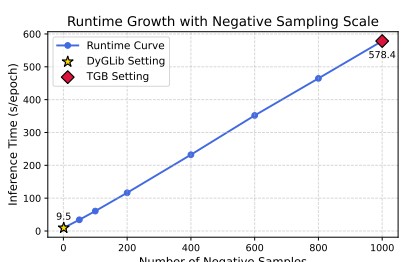

Figure 1: Speed change of TPNet under varying numbers of negative samples on tgbl-wiki (from 1 to 1000).

ity severely limits the practicality of existing methods in real-world applications—such as recommendation, social forecasting, and financial modeling—where over millions of query links must be processed within tight time constraints (Zhou et al., 2022).

To tackle the above issue, this paper makes the following three contributions.

**[Observation]** By analyzing the SOTA method, TPNet, on the TGB Leaderboard, we reveal that the main computational bottleneck lies in converting maintained node memories into edge embeddings, a step that accounts for over $90\%$ of the overall cost (Section 4.2 for details). Interestingly, we find that this step is unnecessary for most query links. By applying a simple threshold-based rule on memory prediction, we can filter out about $80\%$ of the edges, enabling predictions to rely directly on node memories. This simple modification yields up to a 5× speedup while only suffering a negligible performance loss, highlighting the potential of more principled designs.

**[Method]** We further propose a neural selector plug-in, sNet, that adaptively decides whether to rely on maintained node memories or construct more complex edge embeddings for link likelihood computation, thereby transforming a fixed, computationally intensive pipeline into a dynamic and efficient one. A key advantage of sNet is that it requires no additional link prediction model; instead, it directly exploits the hierarchical representations already present in the backbone—node memories and edge embeddings—for adaptive ensembling, which greatly reduces integration overhead and keeps the overall architecture simple. Concretely, sNet adds a prediction head and a selector network. The prediction head outputs link likelihoods using node memories, while the selector evaluates the reliability of these predictions and refines low-confidence cases with likelihoods computed from edge embeddings. To train the non-differentiable selection process, we introduce a tailored surrogate loss together with a dynamic weighting strategy that adaptively balances memory- and embedding-based predictions toward the desired performance target, eliminating the need for extensive trial-and-error tuning.

**[Validation]** Experiments on the TGB benchmark demonstrate that sNet is both effective and efficient. When combined with a state-of-the-art method, it achieves an average $5.56\times$ speedup with only a $0.69\%$ drop in MRR, and even surpasses the original performance on tgbl-comment. Moreover, sNet outperforms the Pareto frontier of the rule-based selector, consistently attaining equal or higher MRR at the same memory fraction. Ablation studies validate the necessity of its submodules, where the proposed training strategy ensures stable, smooth optimization, unlike alternative variants that struggle to simultaneously optimize accuracy and efficiency.

## 2 RELATED WORKS

Temporal link prediction (Qin & Yeung, 2024), which aims to predict future interactions based on historical ones, is a fundamental task in temporal graph learning. Existing approaches can be roughly divided into two categories depending on whether they discretize timestamps: discrete-time methods Sankar et al. (2020); Pareja et al. (2020) and continuous-time methods Kazemi et al. (2020). This paper focuses on the latter because it naturally preserves fine-grained temporal information. For continuous-time methods, early studies mainly sought to learn informative node representations, from which edge logits were decoded via simple transformations (e.g., an MLP) (Kumar et al., 2019; Trivedi et al.; Rossi et al., 2020; Xu et al.; Cong et al., 2023). For example, TGN (Rossi et al., 2020) employ the recurrent neural network to learn node memories and refine them into node embeddings via the graph attention network. However, node-centric representations often overlook correlations between node pairs, which are critical for accurate temporal link prediction (Wang et al., 2021; Yu et al., 2023). To address this limitation, later works shifted toward edge-centric modeling, where node representations are computed conditioned on the target edge, often by constructing relative encodings before generating representations (Wang et al., 2021; Zhang et al., 2024; Yu et al., 2023; Lu et al., 2024). For instance, (Yu et al., 2023) introduce a co-neighbor encoding to inject pairwise context into embeddings, while (Lu et al., 2024) propose a memory-based framework that caches pairwise information in a dynamically updated memory to reduce redundant computation of repeatedly extracted encodings. Despite their effectiveness, these methods typically apply the same per-query likelihood estimation to all edges, which leads to scalability issues when the number of query edges is large. To mitigate this problem, this paper introduces an adaptive prediction framework that dynamically decides whether to use memory-based or embedding-based inference for each query, thereby offering a better tradeoff between effectiveness and efficiency.

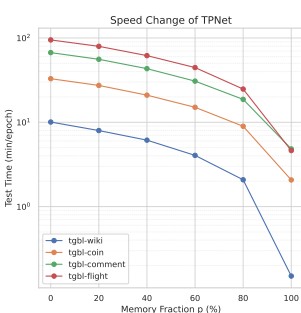 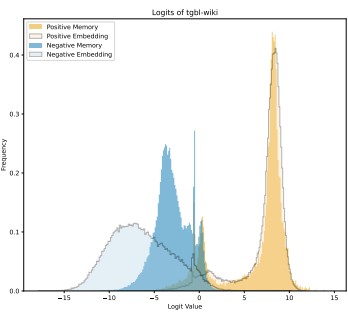 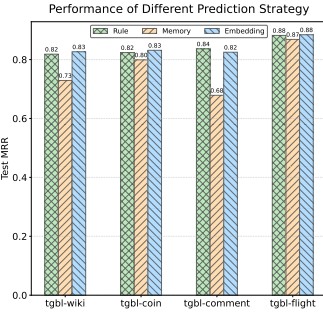

(a) Speed change of TPNet under different mask fraction.

(b) Logit distribution of memories and embeddings.

(c) The final results with a different selection policy.

Figure 2: Performance analysis of memory and embedding predictions.

## 3  PRELIMINARY

**Temporal graph**. We define the temporal graph as a sequence of non-decreasing chronological interactions $\mathcal{G} = [(\{u_1, v_1\}, t_1), (\{u_2, v_2\}, t_2), \cdots]$ with $0 \leq t_1 \leq t_2 \leq \cdots$, where $(\{u_i, v_i\}, t_i)$ indicates an undirected link between $u_i$ and $v_i$ with timestamp $t_i$. Each node $u$ can be associated with node feature $\boldsymbol{x}_u \in \mathbb{R}^{d_N}$, and each interaction $(\{u, v\}, t)$ has link feature $\boldsymbol{e}_{u,v}^t \in \mathbb{R}^{d_E}$, where $d_N$ and $d_E$ denote the dimensions of the node feature and link feature.

**Temporal link prediction.** We formulate temporal link prediction as a ranking problem over potential future edges. Specifically, given a temporal graph $\mathcal{G}_t = [\{u, v, t'\} \mid t' < t]$ observed before $t$, a node $u$, and a candidate set $V_u^t$, the task is to rank the nodes in $V_u^t$ in descending order of their probability of interacting with $u$ at time $t$. Unlike the common formulation that estimates the likelihood $p_{u,v}^t$ of a single future link $(u, v, t)$, our formulation explicitly groups all candidate nodes associated with $u$ at time $t$, aligning more closely with the evaluation protocol used in existing benchmarks (Huang et al., 2023; Yi et al., 2025) and providing a clearer notation for subsequent analysis. Notably, this ranking view is also compatible with the link-likelihood formulation, where the input can be regarded as a set of query links $\{(\{u, v\}, t)\}_{v \in V_u^t}$ that are ranked by their likelihoods $\{p_{u,v}^t\}_{v \in V_u^t}$. In the following, we interchangeably call $p_{u,v}^t$ the likelihood or score for edge $(\{u, v\}, t)$.

## 4  ANALYZE OF SOTA LINK PREDICTION METHOD

The primary goal of this paper is to improve scalability with minimal impact on link prediction performance. To this end, we first examine how the different modules of advanced link prediction methods contribute separately to computational cost and performance. Our analysis focuses on the SOTA method TPNet (Lu et al., 2024), which achieves significant performance improvements over other methods, on the TGB leaderboard[1] (Huang et al., 2023), a widely used large-scale benchmark. In the following, we first provide a brief overview of the TPNet architecture, and then report the performance of its individual modules in terms of effectiveness and efficiency.

### 4.1  COMPONENTS OF THE SOTA METHOD

TPNet belongs to the popular class of memory-based methods (Rossi et al., 2020; Zhang et al., 2023; Su et al., 2024; Jiao et al., 2025), which mainly consist of a memory module and an embedding module. Intuitively, the memory module served as an encoder to compress the node history into memory vectors, which can be considered as a tuple of $(\boldsymbol{M}(t), f_1(\cdot; \boldsymbol{\theta}_1))$. The $\boldsymbol{M}(t) \in \mathbb{R}^{n \times d}$ is the maintained node memory matrix with each row indicating a node's memory at time $t$, and $f_1(\cdot; \boldsymbol{\theta}_1)$ is an updating function, corresponding to update the node memory when a new interaction happens, i.e., $\boldsymbol{M}(t^+) = f_1(\boldsymbol{M}(t), (\{u, v\}, t); \boldsymbol{\theta}_1)$, where $t^+$ indicates the time right after $t$. The superior performance of TPNet largely stems from its specifically designed memory-maintaining mechanism,

---

[1]https://tgb.complexdatalab.com/docs/leader_linkprop/

whose maintained memory matrix provably preserves the pairwise relationships between nodes. For the embedding module, it can be considered as a decoder $f_2(\cdot; \boldsymbol{\theta}_2)$ to refine the node memories into edge embeddings by injecting link prediction-specific information. Formally, given a query link $(\{u, v\}, t)$ it outputs edge embeddings $\boldsymbol{z}_{u,v}^t \in \mathbb{R}^d$ by $\boldsymbol{z}_{u,v}^t = f_2(\mathcal{G}_t, \{u, v\}, \boldsymbol{M}_u(t); \boldsymbol{\theta}_2)$. Then, the link likelihood $p_{u,v}^t$ is computed by feeding the edge embedding $\boldsymbol{z}_{u,v}^t$ into a link predictor $g_2(\cdot; \boldsymbol{\phi}_2)$ (e.g., an MLP). Intuitively, the memory-based architecture introduces two hierarchical representations for each query link $(\{u, v\}, t)$—the edge memory $\boldsymbol{m}_{u,v}^t$ and the edge embedding $\boldsymbol{z}_{u,v}^t$, where the edge memory can be considered as a transformed node memories, i.e., $\boldsymbol{m}_{u,v}^t = \varphi\big(\boldsymbol{M}_u(t), \boldsymbol{M}_v(t)\big) \in \mathbb{R}^d$ with $\varphi(\cdot)$ being operators that convert two representations into one (e.g. concatenation or element-wise product).

## 4.2 Memory Prediction and Embedding Prediction Analysis

In this section, we attach prediction heads (i.e., MLPs that output likelihood) to both edge memories and edge embeddings to compare their efficiency and effectiveness. For the efficiency comparison, we report the runtime when a fraction $p\%$ of edges is randomly assigned to memory predictions, with the remaining $1-p\%$ assigned to embedding predictions. As shown in Figure 2a, fully relying on embedding predictions is at least 10× slower than using memory predictions, and the runtime grows roughly linearly with the proportion of embedding predictions. *This suggests substantial room for efficiency gains if a large fraction of edges can instead be handled through memory predictions.*

Next, we examine the logit distribution of memory and embedding predictions to obtain a more detailed characterization of their properties. Specifically, we collect the edge logits from both memory and embedding predictions and plot the distributions for positive and negative edges, yielding four groups: positive memory, negative memory, positive embedding, and negative embedding. Figure 2b shows that confident memory predictions are typically reliable—edges are almost surely negative when the logit is below –5—and that the overall distributions of memory- and embedding-based logits are largely similar except embedding-based predictions provide clearer separation between positive and negative edges, *suggesting that an adaptive combination of memory and embedding is feasible*. Similar logit distributions for more datasets can be found in Appendix E.1. To further verify the potential of ensembling the memory and embedding predictions, we adopt a rule-based strategy, where we select memory predictions if memory logits are smaller than a predefined threshold, and otherwise, we select embedding predictions. We search the threshold to ensure about $80\%$ of edges are predicted by edge memories, where the rule-based method achieves about a $5\times$ speedup compared to the pure embedding-based one. From Figure 2c, we can observe that TPNet only suffers from a negligible performance drop on the four datasets, and the performance on tgbl-comment even increases, confirming the necessity of ensembling the memory and embedding predictions.

**Conclusion.** From the above analysis, we can conclude that: the prediction flow of memory $\rightarrow$ embedding $\rightarrow$ logit may not be optimal representations for all query links. Instead, an ideal memory-based method should assign each query link to the most suitable representation, which might achieve substantial efficiency gains and even better predictive performance.

## 5 Methodology

Although directly assigning certain edges to memory predictions provides a natural way to ensemble memory and embedding predictions, it remains insufficient since it ignores the interplay between the two. An effective ensemble should exploit this interaction—for example, when an edge is likely to be mispredicted by both memory and embedding models, directing it to the memory branch can still achieve higher efficiency since memory predictions are computationally cheaper. Such decisions, however, cannot be captured by a fixed rule that lacks awareness of joint uncertainty. To model these nuanced relationships and adapt the selection policy to the data distribution, we introduce sNet, a neural selector. In the following, we first describe its architecture and then the training strategy.

### 5.1 Architecture of Neural Selector

**Overall framework.** sNet is designed as a lightweight plug-in that transforms an embedding-only link prediction method into an adaptively memory-embedding hybrid predictor. Specif-

---

**Algorithm 1:** Training Procedure for the Selector Network $\pi(\cdot; \boldsymbol{\psi})$

---

**Input** : Training set $\mathcal{D}_{\text{train}}$, validation set $\mathcal{D}_{\text{val}}$, initial selector parameters $\boldsymbol{\psi}$, initial weight $\lambda_0$, scaling factor $k$, performance threshold $\alpha$, maximum training epoch $n$

**Output:** Trained selector parameters $\boldsymbol{\psi}$

1 Initialize $\lambda \leftarrow \lambda_0$;
2 **for** *epoch* $= 1, 2, \ldots, n$ **do**
3     **for** *each set of query link scores* $\left((s_1^m, \ldots, s_n^m), (s_1^e, \ldots, s_n^e)\right)$ *in* $\mathcal{D}_{train}$ **do**
4         Compute retention probabilities $p_1, \ldots, p_n$ using Equation (1);
5         Compute performance loss $\ell_1$ via Equation (5);
6         Compute efficiency loss $\ell_2 = -\sum_{i=1}^{n} \log p_i$;
7         Form the total loss $\ell = \ell_1 + \lambda \ell_2$;
8         Update $\boldsymbol{\psi} \leftarrow \boldsymbol{\psi} - \nabla_{\boldsymbol{\psi}} \ell$;
9     Computing ranking metric $R$ on the validation set $\mathcal{D}_{\text{val}}$ with the current $\boldsymbol{\psi}$;
10     **if** $R \geq \alpha$ **then**
11         Scale $\lambda \leftarrow \lambda \times k$;
12     **else**
13         Scale $\lambda \leftarrow \lambda / k$;

14 **return** $\boldsymbol{\psi}$

---

ically, let $\mathcal{M} = \{(\boldsymbol{M}(t), f_1(\cdot; \boldsymbol{\theta}_1)), f_2(\cdot; \boldsymbol{\theta_2}), g_2(\cdot; \boldsymbol{\phi}_2)\}$ denote a memory-based link predictor, where $(\boldsymbol{M}(t), f_1(\cdot; \boldsymbol{\theta}_1)), f_2(\cdot; \boldsymbol{\theta_2}), g_2(\cdot; \boldsymbol{\phi}_2)$ indicate the memory module, embedding module, and prediction head on embedding, respectively. sNet can be considered as two modules $\mathcal{S} = \{g_1(\cdot; \boldsymbol{\phi}_1), \pi(\cdot; \boldsymbol{\psi})\}$, where $g_1(\cdot; \boldsymbol{\phi}_1)$ is an additional prediction head on memory representations and $\pi(\cdot; \boldsymbol{\psi})$ is a selector to pick up the memory and embedding predictions adaptively. When a method $\mathcal{M}$ is integrated with the sNet, its link prediction flow will change from $\boldsymbol{M}(t) \xrightarrow{f_2(\cdot; \boldsymbol{\theta_2})} \boldsymbol{z}_{u,v}^t \xrightarrow{g_2(\cdot; \boldsymbol{\phi_2})} p_{u,v}^t$ into a two-stage process: for a given set of query links $(\{u, v_1\}, t), ..., (\{u, v_n\}, t)$, the memory prediction head $g_1(\cdot; \boldsymbol{\phi_1})$ first outputs the corresponding memory scores $s_1^m, s_2^m, \cdots s_n^m \in \mathbb{R}$ based on the edge memories $\boldsymbol{m}_{u,v_1}^t, \cdots, \boldsymbol{m}_{u,v_n}^t$. Then, the selector network $\pi(\cdot; \boldsymbol{\psi})$ outputs retention probabilities $p_1, p_2, \ldots, p_n \in \mathbb{R}$, where $p_i \in [0, 1]$. For each link $(\{u, v_i\}, t)$, if $p_i \leq 0.5$, the memory-based prediction $s_i^m$ is discarded and replaced by a new score $s_i^e$ computed from the embedding prediction head $g_2(\cdot; \boldsymbol{\phi}_2)$. Links are ranked based on their final scores. The above two-stage pipeline is motivated by the unbalanced computational cost of memory and embedding predictions—embedding predictions are far more expensive than memory predictions. We therefore first generate memory predictions and then refine only the unreliable ones with embedding predictions, keeping the number of costly embedding calls to a minimum.

**Design of Selector Network $\pi(\cdot; \boldsymbol{\psi})$.** Section 4.2 shows that the memory score is a reliable indicator of prediction confidence, thereby facilitating the identification of potentially incorrect predictions. However, the same score can carry different significance depending on the query set. For example, consider two sets of memory scores, $(s_1^m, s_2^m, s_3^m) = (0.5, 1, 2)$ and $(s_1^m, s_2^m, s_3^m) = (2, 3, 4)$. In the first set, the score 2 corresponds to the highest one, whereas in the second set, the same score is the lowest, illustrating that the relative importance of a score depends on its context within the query set. Therefore, the selector network models the relative importance of queries within a set, rather than relying on a fixed, threshold-based selection policy. Specifically, the selector network $\pi(\cdot, \boldsymbol{\psi})$ transforms the memory scores $s_1^m, s_2^m, \ldots, s_n^m$ into retention probabilities $p_1, p_2, \ldots, p_n$ as

$$\boldsymbol{h}_i = \pi_1(s_i; \boldsymbol{\psi}_1), \quad p_i = \sigma(\pi_2([\boldsymbol{h}_i, \frac{1}{n}\sum_{i=1}^{n} \boldsymbol{h}_i]; \boldsymbol{\psi}_2)), \tag{1}$$

where $\pi_1(\cdot; \boldsymbol{\psi}_1)$ and $\pi_2(\cdot; \boldsymbol{\psi})$ are two MLPs with $\boldsymbol{\psi}_1, \boldsymbol{\psi}_2$ being learnable parameters, and $\sigma(\cdot)$ is a Sigmoid function. Here, $\pi_1(\cdot; \boldsymbol{\psi}_1)$ first maps each individual score to a feature representation $\boldsymbol{h}_i \in \mathbb{R}^d$, and then $\pi_2(\cdot; \boldsymbol{\psi}_2)$ computes the retention probability $p_i$ based on both the individual representation $\boldsymbol{h}_i$ and the aggregated set representation $\frac{1}{n}\sum_{j=1}^{n} \boldsymbol{h}_j$.

## 5.2 Training Strategy

For a set of query links $(\{u, v_1\}, t), \cdots, (\{u, v_n\}, t)$, the forward process of sNet involves complicated interaction between three different outputs of memory scores $s_1^m, s_2^m, \cdots, s_n^m$, embedding scores $s_1^e, s_2^e, \cdots, s_3^e$ and retention probabilities $p_1, \cdots, p_n$. Jointly optimizing these outputs is challenging, often leading to overfitting and unstable convergence. To address this, we adopt a two-stage training pipeline. In the first stage, we train the link prediction model $\mathcal{M}$ together with the memory prediction head $g_1(\cdot, \boldsymbol{\theta}_1)$. Following standard practice (Yu et al., 2023; Huang et al., 2023; Yi et al., 2025), we use a binary cross-entropy loss, treating each edge as a binary classification task, and optimizing both the memory and embedding scores. In the second stage, we train the selector network $\pi(\cdot, \boldsymbol{\psi})$ on top of the trained link predictor $\mathcal{M}$ and memory head $g_2(\cdot; \boldsymbol{\phi}_2)$, with only $\boldsymbol{\psi}$ being updated. This stage is considerably more efficient than the first, since we can precompute the memory and embedding scores for the query edges and then train only the two MLPs (i.e., $\pi_1(\cdot; \boldsymbol{\psi}_1)$ and $\pi_2(\cdot; \boldsymbol{\psi}_2)$ in Equation (1)) on these scores, thereby avoiding the redundant computation of memory maintenance and embedding generation during training. In this section, we focus on the second stage, and the overall training procedure for this stage is summarized in Algorithm 1.

The training target of the second stage is twofold. First, it aims to maximize ranking performance by promoting positive edges in the ranking. Second, it seeks to retain as many memory predictions as possible, as computing memory scores is significantly less expensive than embedding-based predictions. Accordingly, the training loss consists of the sum of two losses, i.e., $\ell = \ell_1 + \lambda * \ell_2$, where $\ell_1$ corresponds to the performance loss (i.e., minimize the rank of the positive edge), $\ell_2$ corresponds to the efficiency loss (i.e., maximize the fraction of memory predictions), and $\lambda$ is a hyperparameter to control the weight of the two losses. In the following, we first introduce the design of the two loss terms, and then present the dynamic strategy for weight adjustment.

**Design of training loss**. The selection process is non-differentiable: the memory score $s_i^m$ is refined into $s_i^e$ if $p_i \leq 0.5$. To enable training of the selector, this discrete selection must be relaxed into a continuous, differentiable form. For the performance loss $\ell_1$, since the output of $\pi(\cdot, \boldsymbol{\psi})$ is interpreted as the probability of retaining memory score, we adopt a direct idea to maximize the expectation of the rank of the positive links. Formally, for $n$ query links, we treat the memory scores $s_1^m, s_2^m, \ldots, s_n^m$ and embedding scores $s_1^e, \ldots, s_n^e$ as constants. For each query link $i \in \{1, \ldots, n\}$, the final score $s_i$ is then a random variable defined as

$$s_i = \begin{cases} s_i^m, & k_i = 1, \\ s_i^e, & k_i = 0, \end{cases} \quad k_i \sim \text{Bernoulli}(p_i), \tag{2}$$

where $p_i$ is the retention score output by the selector network. Without loss of generality, we assume that the first edge is a positive sample, and the rank of the positive sample can be represented as $r = \sum_{i=2}^n \mathbb{I}[s_1 \leq s_i]$, where $\mathbb{I}[\cdot]$ denotes the indicator function. Then express the expected rank $\mathbb{E}[r]$ as $\sum_{i=2}^n \mathbb{E}[\mathbb{I}[s_1 \leq s_i]]$, we will get

$$\mathbb{E}[r] = \sum_{i=2}^n \left( r_{mm}^i \, p_1 p_i + r_{me}^i \, p_1(1 - p_i) + r_{em}^i \, (1 - p_1)p_i + r_{ee}^i \, (1 - p_1)(1 - p_i) \right), \tag{3}$$

$$r_{mm}^i = \mathbb{I}[s_1^m \leq s_i^m], \quad r_{me}^i = \mathbb{I}[s_1^m \leq s_i^e], \quad r_{em}^i = \mathbb{I}[s_1^e \leq s_i^m], \quad r_{ee}^i = \mathbb{I}[s_1^e \leq s_i^e].$$

Intuitively, Equation (3) decomposes the ranking of the positive edge into a series of pairwise comparisons, i.e., comparing the positive edge scores $s_1^m, s_1^e$ with the negative edge scores $\{s_i^m, s_i^e\}_{i \geq 2}$, where each comparision is associated with a value (e.g., $r_{mm}^i$) and corresponding probability (e.g., $p_1 p_i$). Although $\ell_1 = -\mathbb{E}[r]$ is differentiable with respect to the selector scores $p_1, \ldots, p_n$, directly optimizing it can lead to suboptimal behavior, since it involves products of probabilities (e.g., $p_1 p_i$) computed through the sigmoid layer. Specifically, let $\overline{p}_i$ be the logit of $p_i$ (i.e., $p_i = \sigma(\overline{p}_i)$); then the gradient $\nabla_{\boldsymbol{\psi}} p_i = p_i(1 - p_i)\nabla_{\boldsymbol{\psi}}\overline{p}_i$ becomes very small when $p_i$ approaches 0 or 1. Multiplying two such probabilities (e.g., $p_1 p_i$) further diminishes the gradient, effectively halting optimization (will be shown in Section 6.3). To mitigate this, we further apply the logarithm to (3), following its use in the probability term of the binary cross-entropy loss. Specifically, we first convert (3) into a gradient equally form of $\overline{l}_1 = \sum_{i=1}^n w_i p_i$, where $\nabla_{\boldsymbol{\psi}} - \overline{l}_1 = \nabla_{\boldsymbol{\psi}} - \mathbb{E}[r]$. The weight $w_i$ is defined as

$$w_i = \begin{cases} \text{sg}(p_1) * (r_{mm}^i - r_{me}^i) + (1 - \text{sg}(p_1)) * (r_{em}^i - r_{ee}^i), & i \geq 2 \\ \\ \sum_{i=2}^n \text{sg}(p_i) * (r_{mm}^i - r_{em}^i) + (1 - \text{sg}(p_i)) * (r_{me}^i - r_{ee}^i), & i = 1 \end{cases} \tag{4}$$

where $\text{sg}(\cdot)$ in (4) indicates the stop gradient operation. Equation (4) comes from replacing the product of probabilities (e.g., $p_1 p_i$) with a version that stops the gradient on one factor at a time (e.g., $p_1 \text{sg}(p_i) + \text{sg}(p_1) p_i$) in Equation (3). Then we take the logarithm and sign transforms on the probabilities, which leads to the final loss, i.e.,

$$\ell_1 = \sum_{i=1}^{n} -|w_i| \log(p_i^{\mathbb{I}[w_i > 0]} * (1 - p_i)^{\mathbb{I}[w_i \leq 0]}), \tag{5}$$

where $|\cdot|$ means taking absolute value. Equation (5) will change the term $-w_i \log(p_i)$ into $w_i \log(1 - p_i)$ if $w_i \leq 0$. This is because $p_i$ can make $-w_i \log(p_i)$ approach $-\infty$ by approaching 0 under this situation, potentially affecting the gradients of other terms. For the selection loss $\ell_2$, we simply set it to be $\ell_2 = -\sum_{i=1}^{n} \log(p_i)$, encouraging it to select more memory predictions.

**Dynamic weight adjustment.** The choice of the hyperparameter $\lambda$ requires a prior preference to balance efficiency and effectiveness: a larger $\lambda$ tends to favor efficiency, while a smaller $\lambda$ emphasizes effectiveness. Since the relationship between $\lambda$ and the performance metric of interest is unclear, it often necessitates multiple trials to identify an appropriate value. To this end, we adopt a dynamic adjustment paradigm, where target levels of effectiveness and efficiency are specified in advance, and $\lambda$ is automatically adjusted to meet these requirements, thereby reducing reliance on trial-and-error tuning. Specifically, we set a threshold $\alpha$ for the ranking metric, and hope to maximize the fraction of memory predictions under the condition that ranking performance is no worse than $\alpha$, which is $\max \sum_{i=1}^{n} \mathbb{I}[p_i > 0.5]$, s.t. $R > \alpha$, where R can be any ranking metric of interest (e.g., MRR or NDCG). For the details of the dynamic adjusting rule, we first initialize $\lambda = \lambda_0$ and set a desirable threshold $\alpha$ for ranking performance on the eval set, where $\lambda_0$ and $\alpha$ are two hyperparameters. After each training epoch, if the model's performance on the evaluation set exceeds $\alpha$, we multiply $\lambda$ by $k$; otherwise, we divide $\lambda$ by $k$, where $k > 1$ is a hyperparameter that controls the increasing factor. As we will show in Appendix E.2, this dynamic adjustment is robust to the initialization of $\lambda_0$ and $k$. The full training procedure is shown in Algorithm 1.

# 6 EXPERIMENTAL RESULTS

## 6.1 EXPERIMENTAL SETTINGS

**Dataset and task setting.** We conduct experiments on the TGB benchmark (Huang et al., 2023), which comprises five widely used temporal link prediction datasets: *tgbl-wiki*, *tgbl-review*, *tgbl-coin*, *tgbl-comment*, and *tgbl-flight*. Since the current results for the *tgbl-review* dataset on the TGB leaderboard are based on a buggy version[2], we restrict our experiments to the remaining four datasets. The task settings strictly follow the official TGB protocol: the first 70% of links are used for training, the next 15% for validation, and the final 15% for testing. Negative edges for the training set are randomly sampled, while those for the validation and test sets are provided by the benchmark. Performance is evaluated using Mean Reciprocal Rank (MRR), as computed by the TGB benchmark. Additional details on the datasets and task settings are provided in Appendix B.

**Baselines.** We compare against nine popular temporal link prediction methods: HyperEvent, DyG-Former, TNCN, CTAN, TGN, TGAT, GraphMixer, DyRep, and TPNet. Among them, we integrate sNet into the strongest memory-based approach, TPNet, and refer to the resulting variant as TPNet-sNet. Further details on the baselines are provided in Appendix C.

**Implementation details.** Our code can be found at `https://anonymous.4open.science/r/sNet_submission-96BC`. Implementation details are shown in Appendix D.

## 6.2 PERFORMANCE AND EFFICIENCY COMPARISON

**Baseline comparison.** The performance and runtime are shown in Table 1. Among all methods, TPNet-sNet achieves the second-best overall performance while offering the best efficiency. Notably, TPNet-sNet attains substantial speedups over TPNet with only a negligible drop in MRR, with an average MRR decrease of just 0.69% while delivering an average 5.56× speedup. On

---

[2]Details can be found in this GitHub issue: `https://github.com/shenyangHuang/TGB/issues/95`

Table 1: Test set MRR and runtime of different methods. Speedup is TPNet runtime divided by TPNet-sNet, and Relative MRR is TPNet-sNet relative to TPNet. Top three results are highlighted by **First**, **Second**, and **Third**, with N/A indicating the method is not reported on the TGB Leaderboard.

| Metric | Model | tgbl-wiki | tgbl-coin | tgbl-comment | tgbl-flight | Average |
|---|---|---|---|---|---|---|
| **MRR ↑** | DyRep | 5.00 ± 1.70 | 45.20 ± 4.60 | 28.90 ± 3.30 | 55.60 ± 1.40 | 33.68 |
| | GraphMixer | 11.80 ± 0.20 | N/A | N/A | N/A | N/A |
| | TGAT | 14.10 ± 0.70 | N/A | N/A | N/A | N/A |
| | TGN | 39.60 ± 6.00 | 58.60 ± 3.70 | 37.90 ± 2.10 | 70.50 ± 2.00 | 51.65 |
| | CTAN | 66.80 ± 0.70 | 74.80 ± 0.40 | 67.10 ± 6.70 | N/A | N/A |
| | DyGFormer | 79.80 ± 0.40 | 75.20 ± 0.40 | 67.00 ± 0.10 | N/A | N/A |
| | TNCN | 71.80 ± 0.10 | 76.20 ± 0.40 | 69.70 ± 0.60 | 82.00 ± 0.40 | 74.93 |
| | HyperEvent | 81.00 ± 0.20 | 77.30 ± 0.20 | 76.00 ± 0.20 | 87.70 ± 0.30 | 80.50 |
| | TPNet | 82.65 ± 0.08 | 83.15 ± 0.12 | 82.50 ± 0.55 | 88.43 ± 0.04 | 84.18 |
| | TPNet-sNet | 81.02 ± 0.07 | 82.48 ± 0.21 | 82.78 ± 0.37 | 88.30 ± 0.04 | 83.15 |
| | Relative MRR | 98.03% | 99.19% | 100.34% | 99.85% | 99.35% |
| **Runtime ↓** (min/epoch) | DyRep | 5.82 ± 0.15 | 51.41 ± 1.07 | 123.80 ± 2.04 | 106.79 ± 0.99 | 71.96 |
| | GraphMixer | 34.68 ± 0.17 | N/A | N/A | N/A | N/A |
| | TGAT | 107.90 ± 0.49 | N/A | N/A | N/A | N/A |
| | TGN | 5.62 ± 0.04 | 50.48 ± 0.09 | 127.82 ± 1.99 | 147.49 ± 61.59 | 82.85 |
| | CTAN | 1.51 ± 0.00 | 139.68 ± 1.35 | 275.41 ± 1.06 | N/A | N/A |
| | DyGFormer | 139.26 ± 1.03 | 419.75 ± 0.58 | 847.78 ± 10.23 | N/A | N/A |
| | TNCN | 9.33 ± 0.02 | 37.38 ± 0.14 | 20.65 ± 0.11 | 23.29 ± 0.42 | 22.66 |
| | HyperEvent | 1.83 ± 0.00 | 13.93 ± 0.08 | 30.81 ± 0.71 | 61.89 ± 11.69 | 27.12 |
| | TPNet | 10.13 ± 0.02 | 33.05 ± 0.08 | 67.18 ± 0.24 | 97.73 ± 1.29 | 52.02 |
| | TPNet-sNet | 1.03 ± 0.10 | 7.56 ± 0.67 | 21.52 ± 0.46 | 19.84 ± 0.25 | 12.49 |
| | Speedup | 9.83× | 4.37× | 3.12× | 4.93× | 5.56× |

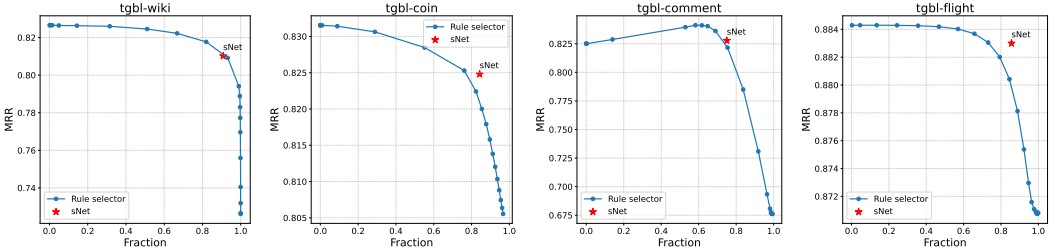

Figure 3: Performance comparison with the rule-based selector, where the Fraction denotes the proportion of memory predictions.

tgbl-comment, TPNet-sNet even surpasses TPNet in MRR. These results highlight the benefits of the dynamic link prediction schema, where the predictor can adaptively select between simple and complex representations for different links, thereby achieving a markedly better balance between performance and efficiency.

**Rule-based selector comparison.** To evaluate the effectiveness of the proposed neural selector, we compare it with a rule-based selector that refines memory predictions into embedding predictions whenever the memory score exceeds a threshold. We plot the performance of the rule-based selector under different thresholds together with sNet in Figure 3. As shown, sNet lies on or even surpasses the Pareto frontier of the rule-based selector, consistently achieving equal or higher MRR at the same Fraction. This demonstrates that the neural selector effectively captures the complex interplay between memory and embedding predictions—something the rule-based approach cannot fully model—thereby attaining a more favorable trade-off between performance and efficiency.

Table 2: Results of different variants for TPNet-sNet, where the Fraction indicates the proportion of memory predictions.

| Method | tgbl-wiki | | tgbl-coin | | tgbl-comment | | tgbl-flight | |
|---|---|---|---|---|---|---|---|---|
| | MRR | Fraction | MRR | Fraction | MRR | Fraction | MRR | Fraction |
| w/o. condition | 81.17 ±0.08 | 91.27 ±0.39 | 83.12 ±0.18 | 10.78 ±18.66 | 83.00 ±0.29 | 70.23 ±0.93 | 88.36 ±0.09 | 43.70 ±3.23 |
| w/o. log | 81.07 ±0.08 | 90.67 ±1.14 | 82.62 ±0.24 | 82.35 ±3.15 | 84.12 ±0.29 | 62.00 ±0.26 | 88.31 ±0.05 | 78.29 ±1.30 |
| w/o. rank loss | 82.66 ±0.08 | 0.00 ±0.00 | 82.70 ±0.03 | 72.20 ±0.46 | 84.12 ±0.29 | 59.39 ±0.10 | 88.43 ±0.04 | 0.00 ±0.00 |
| TPNet-sNet | 81.02 ±0.07 | 90.85 ±1.33 | 82.48 ±0.21 | 84.35 ±2.94 | 82.78 ±0.37 | 75.01 ±0.85 | 88.30 ±0.04 | 85.64 ±0.31 |

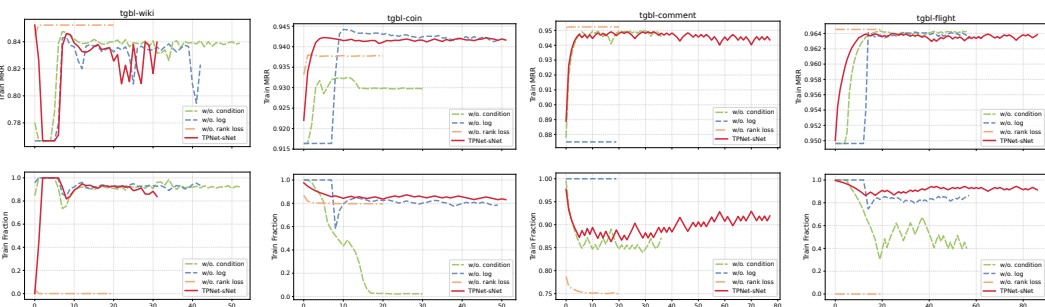

Figure 4: The trajectories of training MRR and Fraction for different methods.

## 6.3 ABLATION STUDY

To verify the effectiveness of the proposed module, we compare TPNet-sNet against the following three variants. 1) **w/o. condition**: set the term $\sum_{i=1}^{n} \boldsymbol{h}_i$ in Equation (1) to zero to verify the necessity of modeling the condition information. 2) **w/o. log**: direct use $-\mathbb{E}[r]$ in Equation (3) as the performance loss $\ell_1$ to verify the importance of taking the logarithm. 3) **w/o. rank loss:** replace the performance loss with the expected memory–embedding loss, i.e., $\ell_1 = \sum_{i=1}^{n} p_i \ell_b(s_i^m, y_i) + (1 - p_i)\ell_b(s_i^e, y_i)$, where $\ell_b$ denotes the binary cross-entropy loss and $y_i$ is the label of the $i$-th query edge ($y_i = 1$ for a positive edge and $y_i = 0$ for a negative edge). From Table 2, we observe that sNet consistently achieves favorable performance, while other variants (i.e., w/o. rank loss and w/o. condition) struggle to optimize both metrics, resulting in highly imbalanced performance. Specifically, w/o. rank loss achieves 0.00 ±0.00 Fraction on tgbl-wiki, while w/o. condition reaches 10.78 ±18.66 Fraction on tgbl-coin. Additionally, w/o. log achieves a similar MRR to TPNet-sNet but with a much lower Fraction; for instance, on tgbl-comment, its Fraction drops from 85.64 to 78.29, while MRR changes only slightly from 88.30 to 88.31. These results confirm the necessity of the proposed modules. To provide a more fine-grained view of how different methods are optimized, we plot the training trajectories of MRR and Fraction in Figure 4. As shown, TPNet-sNet exhibits smoother and more stable trajectories, with MRR and Fraction evolving continuously and fluctuating around stable values. In contrast, other variants either remain largely unchanged, exhibit abrupt jumps, or quickly drop to a trivial state (e.g., Fraction = 0), further highlighting the effectiveness of the proposed loss and dynamic weight adjustment.

## 7 CONCLUSION

In this paper, we investigate the inference scalability of temporal link prediction. Through an analysis of the SOTA method, we highlight that the memory $\rightarrow$ embedding $\rightarrow$ logits prediction flow may not be optimal when both efficiency and effectiveness are considered. Motivated by this insight, we propose a neural selector plug-in, sNet, which enables existing methods to adaptively decide whether to invoke memory or embedding predictions, and is empirically shown to achieve a significantly better trade-off between effectiveness and efficiency on the TGB benchmark. One limitation of our approach is that it focuses solely on two different representations within the same model. Exploring the integration of diverse, heterogeneous representations to further improve performance is a promising direction for future work.

## 8 ETHICS STATEMENT

All experiments were conducted on publicly available datasets, all of which were anonymized. Besides, we carefully examined potential model biases and possible societal impacts. The authors report no conflicts of interest.

## 9 REPRODUCIBILITY STATEMENT

All experimental code and data are released in an anonymous public repository. Using the provided resources, every reported result can be reproduced.

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

Table 3: dataset statistics, where *surprise* denotes the proportion of unseen edges in the test set (a higher value means more edges not seen in the training set), and *negative sample* indicates the number of negative samples per edge in the test set.

| Dataset | Domain | # Nodes | # Edges | # Steps | Surprise | # Negative Sample |
|---------|--------|---------|---------|---------|----------|-------------------|
| tgbl-wiki | interact. | 9,227 | 157,474 | 152,757 | 0.108 | 999 |
| tgbl-coin | transact. | 638,486 | 22,809,486 | 1,295,720 | 0.120 | 20 |
| tgbl-comment | social | 994,790 | 44,314,507 | 30,998,030 | 0.823 | 20 |
| tgbl-flight | traffic | 18,143 | 67,169,570 | 1,385 | 0.024 | 20 |

## A  LLM USAGE

We employed an LLM as a writing assistant to help prepare this manuscript, focusing specifically on improving the clarity and overall readability of the text.

## B  DATASETS AND TASK SETTINGS

The adopted TGB benchmark consists of five datasets, covering a series of real-world application scenarios (Fan et al., 2021; Kumar et al., 2019; Huang et al., 2025; Yu et al., 2024).

- **tgbl-wiki** contains a one-month Wikipedia co-editing network, structured as a bipartite graph of editors and pages. An edge represents a user editing a page at a specific timestamp and includes textual features from that edit. The task is to predict future user-page interactions.

- **tgbl-coin** comprises ERC20 Stablecoin transactions from a period covering the Terra Luna de-pegging event. Modeled as a temporal graph with addresses as nodes and transfers as timestamped edges, the core task is to predict future transaction links between addresses.

- **tgbl-comment** is a directed Reddit reply network from 2005 to 2010, where nodes are users and directed edges represent replies. The task is to predict future replies between users at a given time.

- **tgbl-flight** contains a crowd sourced international flight network from 2019-2022. Airports are modeled as nodes with features for type, continent, ISO region, longitude, and latitude. Edges represent daily flights, with the flight number as the sole edge feature. The task is to predict the existence of a flight between a source and destination airport on a given day.

The statistics of datasets are shown in Table 3

## C  BASELINES

We select the following nine popular baselines:

- **HyperEvent** (Gao et al., 2025) reframes dynamic link prediction as hyper-event recognition by constructing an association sequence with event correlation vectors to capture the structural cohesion among causally related events.

- **DyGFormer** (Yu et al., 2023) is a Transformer-based architecture that employs a neighbor co-occurrence encoding scheme to capture correlations between nodes and a patching technique to efficiently learn from long historical interaction sequences.

- **TNCN** (Zhang et al., 2024) is a temporal version of Neural Common Neighbor (NCN) (Wang et al., 2024), which learns effective pairwise representations for link prediction by dynamically updating a temporal neighbor dictionary and modeling multi-hop common neighbors.

- **CTAN** (Gravina et al., 2024) is a model grounded within the ordinary differential equations (ODE) framework, designed for efficient propagation of information to capture long-range spatio-temporal dependencies in continuous-time dynamic graphs.

- **TGN** (Rossi et al., 2020) utilizes a novel combination of a memory module to store node states over time and graph-based operators to process temporal events.

- **TGAT** (Xu et al.) utilizes a self-attention mechanism to aggregate temporal-topological neighborhood features, coupled with a functional time encoding technique to capture the interaction between time and node features.

- **GraphMixer** (Cong et al., 2023) employs an MLP-based encoder to summarize temporal link information and uses neighbor mean-pooling to aggregate node features, challenging the necessity of more complex recurrent or self-attention mechanisms.

- **DyRep** (Trivedi et al.) models the dynamics of communication and association between nodes using a time-scale dependent multivariate point process to learn evolving low-dimensional node representations.

- **TPNet** (Lu et al., 2024) develops an efficient random feature propagation mechanism to implicitly maintain a novel temporal walk matrix, which incorporates a time decay effect to simultaneously capture both structural and temporal information.

## D  IMPLEMENTATION DETAILS

For sNet, the $\pi_1(\cdot; \psi_1)$ is set to be a one-layer MLP with an output dimension of 128, followed by a ReLU activation. The $\pi_2(\cdot; \psi_2)$ is set to be a four-layer MLP with a hidden dimension of 128, each layer followed by a ReLU activation. We set the initial weight $\lambda_0$ and scaling factor $k$ to be 0.1 and 1.3 respectively. The performance threshold $\alpha$ is set as $0.9 \times \max(R_m, R_e, R_f) + 0.1 \times \min(R_m, R_e, R_f)$, where $R_m$ denotes the performance of pure memory predictions on the validation set, $R_e$ denotes the performance of pure embedding predictions on the evaluation set. $R_f$ denotes the performance of mixed memory–embedding predictions obtained by training sNet with $\lambda$ fixed to 0, which corresponds to optimizing only the ranking objective.

In the first training stage, we use the baseline's official code to train the link predictor model $\mathcal{M}$. After that, we freeze $\mathcal{M}$ and continue training the memory head $g_1(\cdot; \phi_1)$ using the same official implementation. In the second stage, we precompute and store the memory and embedding scores, and then train the two selector MLPs on these scores, where the evaluation and test edges are provided by the TGB benchmark. For the training edges, we select the last $15\%$ of edges relative to the total number of edges in training set, and randomly sample $K$ negative edges for each, where $K$ matches the negative sample number used in the validation and test sets. We adopt the Adam optimizer with a learning rate of $1 \times 10^{-4}$ and apply early stopping with a patience of 20 epochs.

For the baselines, we use the implementations from the TPNet repository, which are inherited from and improved upon DyGLib Yu et al. (2023), a unified library for temporal graph learning. For baselines that are not included in it (i.e., TNCN, HyperEvent, CTAN), we use their official implementation. Experiments are conducted on a Ubuntu server, whose the GPU device is GeForce RTX 3090 GPUs with 24 GB memory. We run each experiment three times and report the average.

## E  MORE EXPERIMENTAL RESULTS

### E.1  LOGIT DISTRIBUTION

Logit distribution of more datasets is shown in Figure 5.

### E.2  HYPERPARAMETERS SENSITIVITY OF THE DYNAMIC WEIGHT ADJUSTMENT

The results with different $\lambda_0$ and $k$ are shown in Table 4, where the MRR are identically the same.

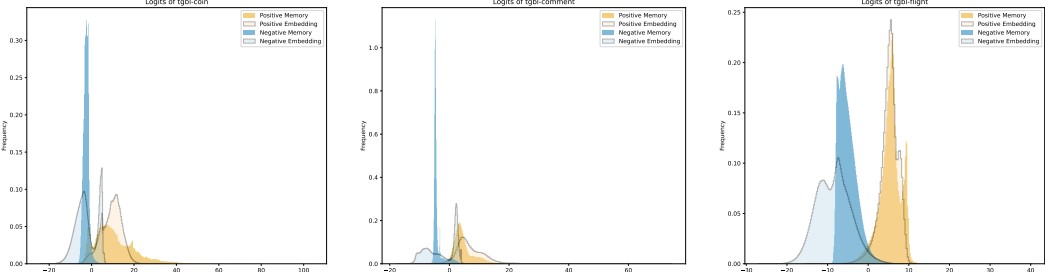

Figure 5: Logit distribution of more datasets.

Table 4: Results of TPNet-sNet with different initial weight $\lambda_0$ and scaling factor $k$ on tgbl-wiki dataset, where Fraction indicates the proportion of memory predictions.

| $\lambda_0$ | $k$ | MRR | Fraction |
|---|---|---|---|
| 0.01 | 1.1 | $81.17 \pm 0.09$ | $88.25 \pm 0.94$ |
| 0.01 | 1.3 | $81.09 \pm 0.01$ | $90.15 \pm 0.78$ |
| 0.01 | 1.5 | $81.02 \pm 0.03$ | $90.40 \pm 1.15$ |
| 0.01 | 1.7 | $81.10 \pm 0.04$ | $90.24 \pm 0.74$ |
| 0.1 | 1.1 | $81.10 \pm 0.05$ | $90.25 \pm 0.84$ |
| 0.1 | 1.3 | $81.03 \pm 0.06$ | $90.64 \pm 1.02$ |
| 0.1 | 1.5 | $81.08 \pm 0.10$ | $90.19 \pm 1.55$ |
| 0.1 | 1.7 | $81.07 \pm 0.05$ | $90.44 \pm 0.87$ |