# OpenReview forum: "sNet: Not All Edges Matter Equally in Temporal Link Prediction"
_ICLR.cc/2026/Conference — Submitted to ICLR 2026_

### Official Review · Reviewer_vnUG · 2025-10-27

**Soundness:** 2
**Presentation:** 1
**Contribution:** 2
**Rating:** 4
**Confidence:** 3

**Summary:**

This study considers the temporal link prediction task on continuous-time dynamic graphs (CTDGs), focusing on the trade-off between quality and efficency. Some empirical pre-experiments showed that converting maintained node memories into edge embeddings dominates the computational cost of existing SOTA methods. A new sNet method was then proposed, which enables a memory-based method to adaptively choose between memory and embedding predictions. Evalution on the well-known large-scale dynamic graph benchmarks (i.e., TGB) further demonstrate the effectivenes of sNet in reaching a significant trade-off between quality and effeciency.

**Strengths:**

**S1**. The proposed method was evaluated on the well-known large-scale dynamic graph benchmarks (e.g., TGB).

**S2**. The key idea of dynamically and adapatively selecting outputs from one out of two modules (i.e., memory and embedding modules) seems interesting.

**S3**. This work anonymously provide its code to ensure reproducibility of experiments.

**Weaknesses:**

**W1**. A better trade-off between inference quality and efficiency is one of the highlighted advantages of sNet. However, there seems no formal analysis about sNet's complexity as well as comparison with those of SOTA baseline methods to theretically analyze its effeciency.


***
**W2**. There are some unclear statements with weak motivations in this paper, which need furthe clarification.

There are no details about how to derive results in Fig. 1. As a result, it is unclear what do the main results in Fig. 1 mean. It is also similar for Fig. 2.

Some key designs of TPNet and sNet were described by lengthy text with many notations and components, which is hard to remember and understand. One limitation of these unclear statements is that it is unclear what do Fig. 2 (a) mean. It is suggested to give some sketch figures or toy running examples to show these key designs.

It seems that one cannot reach the conclsion stated in Section 4.2 that 'runtime grows roughly linearly with the propertion of embedding predictions' based on Fig. 2 (a), i.e., which does not grow linearly espcially for $p \in [80, 100]$.

According to the problem statement in Section 3, it is unclear whether graph attributes (e.g., node and edge attributes) were considered in this paper, as there were no notations to represent these two inputs, while most related methods consider optional attribute inputs.


***
**W3**. There is no pesudo-code to summarize the overall training and inference procedures of sNet. The only pesudo-code (i.e., Algorithm 1) is about one training step (i.e., training of the selector network). As a result, it is still hard to understand how does sNet exactly work by just reading the lengthy descriptions in the main paper.


***
**W4**. Some important experiments are missing, which may not fully validate the effectiveness of some detailed designs in sNet.

There is no parameter analysis about different settings of thereshold $\alpha$.

As the balancing parameter $\lambda$ may vary in different epochs, which leads to different training loss $l$ for different epochs, it is recommended to add a case study about the varied value of $\lambda$ as well as its effect to the convergence of the proposed two-stage training procedure.


****
**W5**. The overall presentation of this paper needs significant improvement.

At the very beginning of this paper (i.e., Section 1), TPNet was first used without any citations.

The font size of Fig. 2, 4, and 5 is too small, which is hard to read.

Fig. 2 was put in page 3 while its corresponding text descriptions are in page 4, which is hard to read with repeated page turning.

One may fail to figure out the distribution of positive embedding in Fig. 2 (b), as there are overlaps between different distributions.

In lines 363-367, there are no citations for the 9 baseline methods.

The overall English presentaion need further polishing. The main paper contain many lengthy pargraphs and thus hard to read and understand, which can be split into shorters sub-paragraphs.

**Questions:**

See **W1**-**W4**.

---

### Official Review · Reviewer_xnNM · 2025-10-29

**Soundness:** 2
**Presentation:** 2
**Contribution:** 3
**Rating:** 4
**Confidence:** 4

**Summary:**

The authors consider a temporal link prediction task and propose a way to speed up making link predictions in practice. They noticed that a large fraction of the runtime for making link predictions is spent on operations for updating link embeddings based on node memories. They argue that, in the majority of cases, these expensive operations are superfluous because predictions can be made from the node memories directly. The authors design a "plugin", which they call sNet, that first generates predictions from node memories, and possibly refines them, using link embeddings, in case the prediction is deemed "unreliable". In an empirical analysis on four TGB benchmark networks, the authors confirm that their approach achieves substantial gains in empirical runtime performance while the prediction performance is nearly unaffected.

I found the paper enjoyable to read, and I thought the authors presented their ideas and goals clearly. I believe the authors consider an important issue with their work, that is, that deep-learning-based methods need to become more runtime-efficient for practical applications. In my opinion, many recent works have focused too much on achieving state-of-the-art performance (bigger numbers in tables) while concerns of practicality were widely neglected.

**Strengths:**

- The authors propose a simple yet effective approach to improve link prediction runtime considerably while prediction performance remains nearly unaffected.
- The paper is well-written and easy to follow. The authors present their ideas clearly.

**Weaknesses:**

- The authors claim that sNet is "empirically shown to achieve a significantly better trade-off between effectiveness and efficiency on the TGB benchmark", however, they do not use any statistical test to support the claim of significance. If the authors wish to claim significance, they should use statistical significance tests. I am also not entirely sure what exactly is supposed to be "significantly better". The trade-off between using the memory of embedding-based predictions? If I understood it correctly, none of the other methods use a combination of these two approaches, so how can the trade-off be "significantly better"?
- It seems as though sNet could be integrated with other TGNNs, however, the authors only consider one method: TPNet. The reason appears to be that, at the time of writing, TPNet is the leader on the TGB leader board. While I don't expect the authors to integrate sNet into other methods, I believe a discussion regarding how this could be done would have strengthened the paper and would have been of interest to practitioners.
- There are relatively many N/A entries in the results in Table 1. From the table caption, I see that this is because these values were not available on the TGB leader board. I believe it would be in order to perform these experiments to fill the table for a more complete picture of how the methods compare.

**Questions:**

I have a couple of questions regarding points that remained unclear to me. I am looking forward to reading the authors' answers to help me clarify those points.

1. It seems as though it should be possible to integrate sNet with other TGNNs, however, this was not really discussed in the paper. Could you briefly discuss whether this is really possible? What are the requirements for integrating your approach with other TGNNs? What could be possible cases when this doesn't work?
2. How was the training done when predictions were generated from node memories instead of edge embeddings? Are you still making as many updates? Specifically, I am wondering whether the training also becomes more computationally efficient because some updates can be skipped, that is, when predictions can be made from node memories instead of edge embeddings. However, to learn when memory-based predictions can be made, I suppose that it is necessary to compare both ways, which would mean that training does not benefit from a potential runtime reduction?
3. You mention that you automatically adjust $\lambda$ in the experiments to reduce reliance on trial-and-error tuning. What does that mean in practice, did you simply make it a learnable parameter? If so, did the training reveal any sort of "rules of thumb" regarding what are good values for $\lambda$?
4. Have you considered what data quality is required for your method to work? For example, how does network density affect sNet? How do false positive or false negative links (incomplete data) affect sNet?
5. There is quite a lot of variability in the achieved runtime speedup in the empirical evaluation. For example, the speedup on tgbl-wiki is $9.83 \times$ while it is "only" $3.12 \times$ on tgbl-comment. Do you know why, or can you speculate why this is the case? And can you generally say something about when we can expect a stronger vs. weaker runtime benefit, for example, depending on network properties?

Minor points
- The fonts in the Figures, especially in Figure 2, should perhaps be a little larger.
- In section 3, "Temporal graph", I believe the "the" should be "a": "We define ~the~ a temporal graph...".
- The heading of section 4 sounds a bit strange, should it perhaps be "Analysis of SOTA Link Prediction Method"? For clarity, you may wish to state the name of that method in the heading, since it seems like it is only one method you consider.
- The text under the ethics statement seems a little out of place and may fit better under a different heading.

---

### Official Review · Reviewer_zPUA · 2025-10-31

**Soundness:** 2
**Presentation:** 3
**Contribution:** 2
**Rating:** 2
**Confidence:** 4

**Summary:**

This paper tackles the inference bottleneck in temporal link prediction on continuous-time dynamic graphs. Profiling the SOTA model TPNet  reveals that over 90% of runtime is spent converting node memories to edge embeddings, though most of these computations are unnecessary. By filtering edges using lightweight memory-based scores, the process can achieve up to 5× speedup with minimal accuracy loss. To address this, the authors introduce sNet, a plug-and-play neural selector that learns to decide which edges require costly recomputation, balancing efficiency and accuracy.

**Strengths:**

1. The paper provides a clear and well-motivated analysis of the computational bottleneck in temporal link prediction, identifying the costly memory-to-embedding transformation and proposing an effective adaptive selection mechanism to reduce redundant computation.

2. The proposed sNet module is general, and can be integrated into existing models, achieving substantial inference speedup with almost no loss in predictive accuracy.

**Weaknesses:**

1. Although the proposed sNet is presented as a general plug-and-play acceleration module for memory-based temporal link prediction models, all empirical results are reported only on TPNet. The paper does not examine whether the same idea maintains its benefit when integrated into other memory-based methods.

2. The motivation of this work heavily relies on the empirical observation that more than 90% of the inference time in TPNet is consumed by the memory-to-embedding transformation stage. However, this profiling analysis is limited to a single implementation and dataset, the conclusion may not hold universally. To make the argument more robust, the authors should provide systematic microbenchmarks across multiple architectures and datasets to confirm that the “embedding branch bottleneck” is a general characteristic of temporal link prediction.

3. The design of the selector relies almost exclusively on the scalar score output of the memory branch to make the selection decision. This formulation implicitly assumes that the reliability of the memory score alone is sufficient to judge whether recomputation is needed. Such an assumption overlooks critical temporal and structural signals, such as time intervals between events, node activity frequencies or historical edge recency, which strongly correlate with representation staleness and prediction difficulty. As a result, the selector may fail to capture nuanced temporal uncertainty, especially in highly dynamic graphs where recent or rare interactions are underrepresented in memory states.

4. The sNet architecture effectively learns a binary gating policy to switch between two scoring functions. However, the paper does not provide a theoretical justification for how this discrete selection boundary should be shaped in relation to the score distribution, nor how it interacts with ranking metrics such as MRR. The current training framework uses a heuristic differentiable relaxation to approximate this binary process, but without analyzing its expected error bound or bias toward one branch.

5. The proposed surrogate ranking loss lacks rigorous theoretical justification. The loss approximates the expected ranking quality under stochastic gating through a log-transformed expectation, but this approximation does not align precisely with standard ranking metrics such as MRR and provides no formal guarantee on the approximation error. Moreover, the derivation relies on an implicit pairwise independence assumption, neglecting the global dependencies that characterize true ranking objectives. Without a comparative analysis against established differentiable ranking surrogates (e.g., SoftRank, LambdaRank), it remains unclear whether the proposed surrogate offers any tangible advantage in optimization stability, convergence speed, or fidelity to the target ranking metric.

6. The dynamic balancing mechanism between accuracy and efficiency depends on two critical hyperparameters: the target performance threshold α and the dynamic weight λ. Although the authors propose a heuristic update rule for λ, the initialization value λ₀ and update multiplier k remain manually tuned. The sensitivity of performance with respect to these parameters is only partially explored on a single dataset. The author should include a more systematic sensitivity analysis across datasets, report convergence behavior  and discuss potential failure cases when α is set too aggressively.

**Questions:**

Please see the weakness above.

---

### Official Review · Reviewer_yZ3b · 2025-10-31

**Soundness:** 2
**Presentation:** 3
**Contribution:** 2
**Rating:** 4
**Confidence:** 3

**Summary:**

This paper addresses the problem of low inference efficiency in temporal link prediction and proposes a plug-and-play neural selector module called sNet. sNet adaptively decides between memory-based and embedding-based predictions, achieving a significant improvement in computational efficiency while maintaining predictive accuracy. Experimental results on the TGB benchmark demonstrate the effectiveness and practical value of the proposed method in temporal graph modeling.

**Strengths:**

1. The paper has a clear and well-motivated problem statement, focusing on a specific and practically relevant issue in temporal link prediction.
2. The proposed module sNet is simple, and easy to integrate into existing memory-based models without structural modification.
3. The proposed framework shows high practical relevance and can potentially benefit real-world applications such as social networks, recommender systems, and financial risk modeling.

**Weaknesses:**

1. Section 4 provides background on the TPNet model and its computational bottlenecks, but the analysis is largely descriptive and could be condensed or moved to the Related Work or Introduction sections as part of the motivation discussion.
2. The evaluation metrics are limited to runtime and MRR. Other important aspects like energy efficiency are not reported, which restricts the comprehensiveness and practical interpretability of the results.
3. Although sNet is described as a plug-and-play module, it is only evaluated on TPNet, without testing on other architectures. This limits the evidence for its generality and cross-architecture applicability.

**Questions:**

Please see the weaknesses.

---

### Meta-Review · Area_Chair_LW6v · 2026-01-01

**Summary:**

The paper targets inference inefficiency in temporal link prediction on continuous-time dynamic graphs. The authors noticed that a large fraction of the runtime for making link predictions is spent on operations for updating link embeddings based on node memories. To address this, it proposes sNet, a plug-and-play selector that adaptively decides between memory-based and embedding-based predictions, achieving a significant improvement in computational efficiency. On TGB benchmarks, the authors report substantial speedups with little MRR loss, arguing sNet offers a better quality–efficiency trade-off.


The authors did not participate in the rebuttal. None of the concerns from the reviewers is resolved.

**Reviewer Concerns:**

The authors did not participate in the rebuttal. Here is a summary of key concerns from the reviewers:

1/ Despite sNet is claimed as “plug-and-play,” results are only on TPNet. (yZ3b, zPUA, xnNM)

2/ The evaluation metrics are limited to runtime and MRR. Some important experiments are missing. Reviewers request energy/complexity analysis and statistical significance tests for “significant” claims and ablation studies. (yZ3b, xnNM, vnUG)

3/ The surrogate ranking loss and gating relaxation lack theoretical justification and comparison to established differentiable ranking surrogates. (zPUA)

4/ The accuracy–efficiency controller (target threshold α, dynamic weight λ) were heuristically tuned. Its sensitivity and convergence behavior are only partially explored. (zPUA, vnUG)

**Reviewer Scores:**

All reviewers gave negative scores. There are no rebuttal discussions.

---

### Decision · Program_Chairs · 2026-01-26

Reject